# WHEN MACHINES WRITE: A METHOD FOR DETECTING AI-EDITED TEXT

## ABSTRACT

Existing AI-text detectors have reported great success in detecting AI-generated content created by text completion and question answering. We consider a more challenging problem—distinguishing between human-written content and human-written, AI-edited content (hwAI text), in which the signals are weaker and existing methods are less satisfying. We propose *word-list-assisted prompting* as a new method. It is based on two empirical observations: (i) Word-count features, despite being sparse, are powerful in detecting hwAI text. (ii) The direct prompting approach, though conventionally not recommended, becomes effective after being supplied a selected word list in the prompt. To this end, we develop two feature selection methods, leveraging the advancement in large-scale multiple testing and topic modeling. Our prompting approach, powered by these feature selection methods, achieves appealing performance in detecting hwAI text in several data sets containing academic abstracts, movie reviews, news, and student essays.

## 1 INTRODUCTION

The rapid advancement of artificial intelligence motivates the problem of detecting AI-generated text, to help prevent harmful misinformation (Kreps et al., 2022) or incorrect advice in AI-generated content. Many methods have been proposed (Solaiman et al., 2019; Gehrmann et al., 2019; Liu et al., 2019; Mitchell et al., 2023; Bao et al.; Hans et al., 2024; Guo et al., 2024a; Tian et al.; Zhang et al.; Yang et al.; Mao et al.; Wang et al.; Chakraborty et al., 2024; Verma et al., 2024; Guo et al., 2024b), and the performance of state-of-art methods on various benchmark data sets has been appealing. However, the AI-generated text in most studies were created from either text completion or question answering. In the former, a small number of initial tokens of human-written text were fed into a large language model (LLM) to generate the remaining text. In the latter, a question was given to an LLM to generate the answer. The discriminative signals in these settings were relatively strong.

However, in many situations, the content to examine involves human-AI collaboration. Two examples are the *human-written & AI-edited* text and the *AI-written & human-edited* text. How to distinguish them from the *purely human-written* text and the *purely AI-generated* text is a problem of great interest. For example, many journals and conferences now require authors to disclose their use of AI, including polishing writing (Liang et al., 2023; 2024). An automatic detector of AI-editing provides a double-check in addition to author self declaration. As another example, a public article drafted by AI and then edited by human is likely to be less harmful than an article purely generated by AI without human input. The first example above is about distinguishing between *human-written* text and *human-written & AI-edited* text, and the second is about distinguishing between *AI-written* text and *AI-written & human-edited* text. A common feature in these two problems is that the text from two classes are quite similar and the discriminative signals are much weaker (Hashemi et al., 2024; Tao et al., 2024; Wang et al.) than in the standard setting in the literature. Our experiments (see Section 3) show that popular detectors have less satisfying performance in such cases.

In this paper, we develop a new classifier particularly suitable for such weak discriminative signals. We mainly focus on distinguishing between *human-written* text and *human-written & AI-edited* text, because it is cheap to obtain such data by using commercial LLMs.[1] Our method differs from all

---

[1]Our method can also be directly applied to distinguishing between *AI-written* text and *AI-written & human-edited* text, simply by changing the input data. However, since obtaining human-edited text is expensive, we do not include such experiments in this paper.

Figure 1: The comparison between a human-written text abstract (left) and its AI-edited version (right). Both abstracts are abbreviated with [...] to save space. The figure is generated by *diffchecker.com*.

existing methods in two aspects: First, we utilize multiple testing and topic modeling techniques to select a word list that has discriminative power. This step explores intrinsic sparsity in our problem to harness maximum power from weak discriminative signals. Second, we introduce a new prompting technique that utilizes the selected word list to obtain a classification decision. Since this step prompts an LLM, it will capture the higher-order signals beyond bag-of-words features. We demonstrate that our method is tuning-free, computationally much faster than many existing methods, and significantly outperforms several strong baselines.

**The hwAI-text detection problem, our method, and experiment designs:** Given an academic abstract or a product review, we prompt an LLM to modify it. Such content is harder to detect than purely AI-written content. For brevity, we call it *hwAI-text detection*. Existing AI-text detection tools do not have satisfactory performance in detecting hwAI text. For example, when we apply RoBERTa (Liu et al., 2019) and DetectGPT (Mitchell et al., 2023) to AI-edited academic abstracts, the accuracy is always below 80%, sometimes even much lower (but our method consistently achieves an accuracy above 90%). To see why the hwAI-text detection is a challenging problem, we show in Figure 1 a human-written abstract (Kerman & Gelman, 2007) and its revision by GPT-4o-mini with the prompt: *"Given the following abstract, make some revisions. Make sure not to change the length too much."* We observe that the edited version uses many identical words as the original one, leading to weak signals.

We propose a new solution to hwAI-text detection, based on two empirical observations.

*Observation 1: Bag-of-word features alone capture a lot of discriminative signals. Furthermore, exploring sparsity in such features can significantly prevent over-fitting.* The example in Figure 1 suggests that most edit by AI is word replacement. For instance, "involves manipulating and summarizing" is changed to "encompasses the manipulation and summarization". Such difference can be revealed in word counts. [2] Since useful word-count features are very sparse, we can apply sparse feature selection methods from traditional statistics and machine learning. Due to the simplicity of bag-of-word models and the focus on sparsity, these methods are resistant to over-fitting and can perform well even when training data are not homogeneous as testing data.

*Observation 2: The direct-prompting approach (asking an LLM to distinguish human- and AI-written text) becomes very effective after being supplied bag-of-word features.* In the literature, the direct prompting approach was conventionally not recommended for AI-text detection (Bhattacharjee & Liu, 2024; Huang et al., 2025). However, we discover surprisingly that the prompting approach can be effective in detecting hwAI-generated text, as long as we include in the prompt a list of selected words that have discriminative power. For example, in one of our experiments (see Table 5), directly prompting GPT yields an error rate of over 50%, but after we incorporate a selected list of words by HC (to be introduced), the error rate immediately drops to less than 10%.

Motivated by the above observations, we propose the *word-list-assisted prompting* approach. It takes a training data set (which size needs not be large) and applies a word-count-based feature selection method. The selected word list, together with the text to classify, is used to prompt an LLM to get

---

[2]We consider the counts of natural words, without stemming and lemmatization. This is one of the keys for the success of our method. In fact, the AI edits in our experiments also involve operations such as token reordering, paraphrasing, or sentence splitting. Fortunately, these operations can still be reflected in the counts of natural words. For example, in the example in Figure 1, *"we illustrate the use of this new programming environment with examples of Bayesian computing, demonstrating miss-value imputation"* is re-ordered to *"to exemplify this new programming environment, we present examples of Bayesian computations that demonstrate missing-value imputation."* Here, *examples* and *demonstrating* appear in the human-written abstract, while *exemplify* and *demonstrate* appear in the AI-edited version

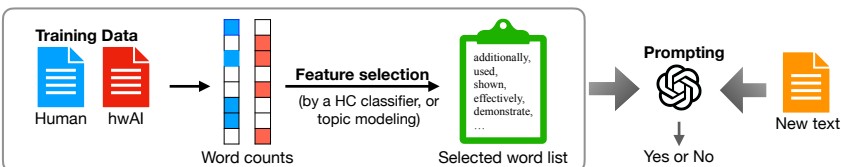

Figure 2: An illustration of our proposed method.

an answer (see Figure 2). To pass more information to the LLM, we use a *signed* word list in the prompt, where each word is marked to be preferred by either human or AI.

The remaining question is to find a good feature selection method. We hope that it is easy to compute, requires no tuning, and takes advantage of feature sparsity. We propose two methods. The first is for the case where training data are from the same domain and relatively homogeneous. We assume that text documents in each class (human or hwAI) are i.i.d. from a discrete distribution on the vocabulary and test whether the two distributions are equal (Balakrishnan & Wasserman, 2018; Cai et al., 2024). We select features by first computing a p-value for each word and then using Higher Criticism (HC) Donoho & Jin (2004; 2008), a statistical testing framework for rare and weak signals, to obtain a data-driven threshold on p-values. This method is entirely tuning-free. The second method assumes that text documents in each class are drawn from mixtures of $K$ discrete distributions—a topic model Blei et al. (2003); Ke et al. (2024). We first adapt the algorithm in Ke & Wang (2024) to estimate the topic model for each class and then select the words whose topic loadings are most different between two classes. This method has only two tuning parameters, which can be chosen by cross validation.

In our experiments, we tried both (i) training a linear classifier on selected word-count features and (ii) using selected words in a prompt as in Figure 2. The first one already achieves good performance in hwAI-text detection and sometimes even outperforms state-of-art algorithms. But (ii) is consistently better than (i). These results confirm that word counts capture *most-but-not-all* discriminative signals. The combination of word selection and prompting turns out to be a satisfying solution.

Since our method requires training data, it is interesting to investigate how many training samples are sufficient and whether our method permits heterogeneity between training and testing. Among the data sets in our experiments, two (academic abstracts (Ke et al., 2024) and movie reviews) have the author information. We propose three experiment designs: PAD (pooling-author design), where both the training and testing data come from many authors, CAD (cross-author design), where the training data are from one author and the testing data are from another, and SAD (same-author design), where the training and testing data are from the same author. PAD is a standard design, and CAD and SAD involve small training sample size (often less than 25 text documents). CAD is more challenging than SAD, because the writing styles of different authors can be significantly different, leading to heterogeneity between training and testing data. We find that our method works for all three designs.

**Comparison with the literature:** DetectGPT Mitchell et al. (2023) is a popular method for AI-text detection. It uses a novel perturbation-evaluation framework, which relies on a key insight: perturbations of AI-generated text tend to decrease the log probability, while perturbations of human-written text may increase or decrease the log probability. Besides the log-probability, other statistical metrics such as entropy and probability rank (Gehrmann et al., 2019) have also been used. Many variants of the DetectGPT framework have been proposed. Fast DetectGPT (Bao et al.) modifies the perturbation step from mask-filling to a fast sampling procedure; DNA-GPT (Yang et al.) perturbs text by feeding the first $\gamma$ fraction of tokens and asking the source LLM to generate the remaining tokens; Raidar (Mao et al.) perturbs text with 're-writing'; and Binoculars (Hans et al., 2024) adopts new evaluation metrics utilizing the cross-perplexity between two LLMs.

While perturbation/rewriting is used as a tool in such methods, the original text to classify is still purely AI-generated. However, in our problem, the AI-generated content is by itself a minor perturbation of human-written content; and DetectGPT and its variants become unsatisfactory. Other disadvantages include the need to compute log-probabilities through an API and the lack of an explicit threshold on evaluation metrics (hence, these methods still do not directly output a classification decision). In comparison, our method is crafted for detecting AI-edited text, leveraging statistical techniques of sparse feature selection and tuning-free threshold choice; and our method interacts with an LLM only

through prompting, without the need to compute log-probabilities. On the other hand, DetectGPT is a zero-shot detector, while our method requires a small training sample.

Fine-tuning a neural-network-based classifier is another approach for AI-text detection. For example, Uchendu et al. (2020) considered the RNN and CNN architecture; Tay et al. (2020) used transformers; Zellers et al. (2019) fine-tuned the GROVER model; and Ippolito et al. (2020) fine-tuned the BERT model. The fine-tuned RoBERTa Liu et al. (2019); Solaiman et al. (2019); Fagni et al. (2021); Adelani et al. (2020) is most popular within this class. Other methods include Badaskar et al. (2008); Tian et al.; Zhang et al.; Guo et al. (2024a), which also aim to train complex classifiers. These methods require a large training sample. In contrast, our training step only uses word counts and can work well with a small training sample (especially by leveraging sparsity). Although word-count features are conventionally thought to be less informative than neural-network features, they indeed work very well for our problem, likely due to the nature of AI-editing. Recent work has also focused on improved baselines for AI-text detection. Verma et al. (2024) introduced Ghostbuster, which uses structured n-gram features and probability features from multiple models.

The direct prompting approach is generally not recommended for AI-text detection (e.g., Bhattacharjee & Liu (2024) found that ChatGPT struggled to identify AI-generated text), but we make a surprising discovery that the word-list-assisted prompting can dramatically improve it. This draws an interesting connection to the recent studies of advanced prompting strategies, such as Chain-of-Thought (CoT) prompting Wei et al. (2022), Self-Consistency prompting Wang et al. (2023), and Tree-of-Thoughts Yao et al. (2023). Our prompting strategy is different: Essentially, we incorporate a summary/description of a separately trained (shallow) model. This prompting strategy may be useful for other prediction tasks. So far, we have only used the form of a word list. Possible extensions could be incorporating an n-gram list.

Our work is also connected to the statistical literature about authorship attribution (Mosteller & Wallace, 1963; Kipnis, 2022; Cai et al., 2024). All of them used the bag-of-word models, but our method involves prompting an LLM. In our problem, if we merely use word-count features to build a classifier, it cannot achieve the best accuracy. Additionally, we propose new ideas for word-count feature selection. For example, the equal-weight topic modeling (see Section 2.2) for paired documents is new and has never been proposed in the literature.

## 2 THE PROMPT AND TWO FEATURE SELECTION METHODS

The prompt contains both a signed word list and the test document. Since the word list is generated by feature selection and varies across experiments, we use Python code to generate the prompt, as shown in Figure 3, where `human_words` and `ai_words` are two string lists (see Table 3 for example), and `text` is the testing document. We may change `ChatGPT` to a different LLM name, but this does not have much impact on the final classification results.

```
prompt = f"""You are an expert forensic linguist.

Task ──────────────────────────────
Decide whether the following document was
• written entirely by a human        → label "human"
• Written by a human but then edited or rewritten by ChatGPT  → label "ChatGPT"

Prior knowledge ───────────────────
A trusted statistical detector (HC) has already analysed the text:
Discriminative vocabulary discovered by PHC:
• Common in human documents    :  {', '.join(human_words)}
• Common in documents edited by ChatGPT:  {', '.join(ai_words)}

Answer with exatcly the following format without outputing your reasoning.
prediction:<label>
where <label> is either human or ChatGPT (case-sensitive).
Document ───────────────────────────
{text}
──────────────────────────────────
"""
```

Figure 3: The python code for prompt generation.

We then develop two feature selection algorithms for obtaining `human_words` and `ai_words` from training data. They are both easy to implement and require no or little tuning.

### 2.1 THE HIGHER CRITICISM (HC) METHOD FOR FEATURE SELECTION

Suppose we have $n_h$ human-written documents and $n_{ai}$ hwAI-written documents. Let $X_{j,i}^h$ be the count of word $j$ in human-written document $i$, and let $X_{j,i}^{ai}$ be the count of word $j$ in hwAI-written

document $i$. We model these counts as Poisson variables. For $1 \leq j \leq p$ ($p$ is the vocabulary size),

$$
\begin{cases}
X_{j,i}^{h} \stackrel{iid}{\sim} \text{Poisson}(\lambda_j), & 1 \leq i \leq n_h, \\
X_{j,i}^{ai} \stackrel{iid}{\sim} \text{Poisson}(\mu_j), & 1 \leq i \leq n_{ai}.
\end{cases}
\tag{1}
$$

Let $\delta_j = |\lambda_j - \mu_j|$, $1 \leq j \leq p$. A word is considered discriminative if $\delta_j \neq 0$. Presumably, for only a small fraction of $j$, $\delta_j > 0$, and each nonzero $\delta_j$ is relatively small, so the *signals are Rare/Weak*.

For each word, we compute a $t$-statistic comparing its frequencies in human- and hwAI-texts:

$$
z_j = (\bar{X}_j - \bar{Y}_j)/\sqrt{s_{Xj}^2/n_h + s_{Yj}^2/n_{ai}},
\tag{2}
$$

where $X_j$ and $s_{Xj}^2$ are the empirical mean and variance of $\{X_{j,i}^h : 1 \leq i \leq n_h\}$, respectively, and $Y_j$ and $s_{Yj}^2$ are defined similarly for $\{X_{j,i}^{ai} : 1 \leq i \leq n_{ai}\}$. When $j$ is a non-discriminative word, as its total count (in all training documents) tends to infinity, it can be shown that $z_j \to N(0,1)$ in law, so we can approximate the $p$-value by $\pi_j = 2\Phi(|z_j|) - 1$, where $\Phi$ is the CDF of $N(0,1)$.

We first threshold $\pi_j$ to obtain the list of discriminative words and then use the sign of $z_j$ as the sign for each selected word. This gives the signed word list.

Despite that this approach is simple, there is a critical question: how to select a data-driven threshold for p-values? Cross-validation (CV) is not ideal, especially when the training sample size is small, where the CV-threshold is often unstable. False discover rate (FDR) control faces another issue: It requires the user to choose a target FDR level, which becomes another tuning parameter by itself.

In the statistical literature, Higher Criticism (HC) Donoho & Jin (2004; 2008; 2009); Jin (2009); Jin & Wang (2016) is a convenient approach to data-driven threshold selection, and it has been successfully applied in large-scale multiple testing and high-dimensional sparse classification and clustering. We apply HC to our problem as follows: Sort the p-values and let $\pi_{(m)}$ denote the $m$th smallest value among $\pi_1, \pi_2, \ldots, \pi_p$. Compute

$$
m^* = \underset{1 \leq m \leq p}{\arg\max}\{HC_{p,m}\}, \qquad HC_{p,m} = \frac{\sqrt{p}[m/p - \pi_{(m)}]}{\sqrt{(m/p)(1 - m/p)}}, \qquad 1 \leq m \leq p.
\tag{3}
$$

To explain (3), let $N_\alpha$ be the the number of significant p-values under a threshold $\alpha$. When all words are non-discriminative, p-values are uniformly distributed in $[0, 1]$, so that $N_\alpha \sim \text{Binomial}(p, \alpha)$. As a result, $U_\alpha := \frac{\sqrt{p}(N_\alpha/p - \alpha)}{\sqrt{(N_\alpha/p)(1 - N_\alpha/p)}}$ is a properly-scaled t-statistic. Meanwhile, when $\alpha = \pi_{(m)}$, the number of significant p-values is exactly $m/p$. Then, $HC_{p,m}$ is nothing but $U_\alpha$ evaluated at $\alpha = \pi_{(m)}$. To this end, $HC_{p,m}$ measures the evidence of rejecting the global null hypothesis when we only look at the $m$ smallest p-values. We choose $m^*$, at which this evidence is the strongest. Let

$$
\hat{S} = \{1 \leq j \leq p : \pi_j \leq \pi_{(m^*)}\}.
\tag{4}
$$

We further divide $\hat{S} = \hat{S}_1 \cup \hat{S}_2$, where a word in $\hat{S}_1$ (and in $\hat{S}_2$) has a positive (negative) $z$-score; see (2). These are the word lists to insert into the prompt. The whole method is entirely tuning-free.

**Remark 1**: HC was originally proposed in Donoho & Jin (2004) for large-scale multiple testing and shown to achieve an optimal phase diagram (Donoho & Jin, 2004). In Donoho & Jin (2008; 2009); Jin (2009); Jin & Wang (2016), HC was used in high-dimensional classification when useful features are Rare/Weak, and it was shown to have optimal performance. In our problem, *useful features are rare*: Despite a large vocabulary, only a small fraction of words show meaningful differences between human and AI usage. This reflects the fact that modern language models are trained to mimic human writing patterns for most common words. Additionally, *useful features are weak*: When differences in word usage exist, they are typically small relative to the natural variation in text, and each individual useful word only contributes weakly to the classification decision (but they will act collectively to enhance classification). HC is particularly well-suited for our problem.

**Remark 2**: After having $\hat{S}_1$ and $\hat{S}_2$, we can also run a linear classifier. Let $d \in \mathbb{R}^p$ be the word count vector of a test document. Let $u_j^{test} = d_j - (n_h \bar{X}_j + n_{ai} \bar{Y}_j)/(n_h + n_{ai})$ be the centerdized word count, $1 \leq j \leq p$ (notations are the same as those in (2)). Compute $L(d) = (\sum_{j \in \hat{S}_1} u_j^{test}) - (\sum_{j \in \hat{S}_2} u_j^{test})$. This classifier predicts 'human' if $L(d) \geq 0$ and 'hwAI' otherwise. We call it the HC classifier, and call the prompting approach that incorporates $\hat{S}_1$ and $\hat{S}_2$ the HC-LLM classifier.

## 2.2 THE TOPIC MODELING METHOD FOR FEATURE SELECTION

In (1), we assume that the distribution of word counts is the same for all documents by human (or hwAI). This assumption can be restrictive in practice. We replace it by a more realistic model, the topic model (Hofmann, 1999; Blei et al., 2003; Ke et al., 2024). Suppose that we have $n$ documents written on a vocabulary of $p$ words, discussing $K$ different topics. For each $1 \leq k \leq K$, we have a topic vector $A_k \in \mathbb{R}^p$, which is a probability mass function (PMF) defined on the vocabulary. Also, for each $1 \leq i \leq n$, let $X_i \in \mathbb{R}^p$ be the word count vector for document $i$, and let $N_i$ denote the document length. Suppose that each document $i$ is associated with a topic weight vector $w_i \in \mathbb{R}^K$, where $w_i(k)$ is the fractional weight that document $i$ puts on topic $k$. We model $X_i$ by $X_i \sim \text{Multinomial}(N_i, \Omega_i)$, with $\Omega_i = \sum_{k=1}^K w_i(k) A_k$. Topic modeling aims to estimate $A = [A_1, A_2, \ldots, A_K]$ using $X_i$'s. There are many topic modeling algorithms. We mainly use the fast spectral algorithm Topic-SCORE (Ke & Wang, 2024).

We assume that there are $K$ topics in both human-written documents and hwAI-written documents, and the two sets of topics have one-to-one correspondence. Write $A^h = [A_1^h, \ldots, A_K^h]$ and $A^{ai} = [A_1^{ai}, \ldots, A_K^{ai}]$. Let $e_1, e_2, \ldots, e_p \in \mathbb{R}^p$ be the standard basis vectors. We assume

$$e_j'(A^h - A^{ai}) \text{ is a nonzero vector only for a small fraction of words.} \tag{5}$$

Let $\widehat{A}^h$ and $\widehat{A}^{ai}$ be the estimated topic matrices by applying Topic-SCORE to two classes of training documents separately. Since the estimated topic matrix is subject to an arbitrary column permutation, we search for a permutation $\tau(\cdot)$ to minimize $\sum_{k=1}^K \|\widehat{A}_k^h - \widehat{A}_{\tau(k)}^{ai}\|_1$. After the topics are well-aligned, we compute a statistic $T_j$ for each word as below. For a threshold $t > 0$ to be determined, we select only words such that the difference is bigger than $t$:

$$\widehat{S}(t) = \{1 \leq j \leq p : T_j \geq t\}, \qquad T_j = \|e_j'(\widehat{A}^h - \widehat{A}^{ai})\|_1, \qquad 1 \leq j \leq p. \tag{6}$$

We further divide $\widehat{S}(t)$ into two subsets: $\widehat{S}_1(t)$ contains those words where $\sum_{k=1}^K e_j'(\widehat{A}^h - \widehat{A}^{ai}) \geq 0$ and $\widehat{S}_2(t)$ contains those words where $\sum_{k=1}^K e_j'(\widehat{A}^h - \widehat{A}^{ai}) < 0$. This method has tuning parameters $(K, t)$. Unlike in Section 2.1, we don't have a perfect data-driven threshold choice here. In our experiments, we use cross-validation to choose $(K, t)$ (see Section 3).

**Variant: Equal-weight Topic Modeling.** In most of our experiments, the training documents are one-to-one paired: For each human-written document, we have an AI-edited version. To take advantage of the pairing information, we propose a *equal-weight topic model*. Suppose $n_h = n_{ai} = n$. Let $X_i^h$ and $X_i^{ai}$ be the word count vectors for the $i$th pair. We assume that $X_i^h$ satisfies the topic model with its own $(N_i^h, \Omega_i^h, A^h, w_i^h)$, and $X_i^{ai}$ also satisfies the topic model with $(N_i^{ai}, \Omega_i^{ai}, A^{ai}, w_i^{ai})$, and $w_i^h = w_i^{ai} = w_i$, for $1 \leq i \leq n$. We adapt Topic-SCORE to estimate $(A^h, A^{ai})$ under this constraint. Our method is based on a key observation: Recall that $\Omega_i^h$ and $\Omega_i^{ai}$ are the population word frequency vectors in a pair of documents, respectively. If we stack them together into a $2p$-dimensional vector, we obtain that $\begin{bmatrix} \Omega_i^h \\ \Omega_i^{ai} \end{bmatrix} = \sum_{k=1}^K w_i(k) \begin{bmatrix} A_k^h \\ A_k^{ai} \end{bmatrix}$, for $1 \leq i \leq n$. This is a structure similar to that in the topic model, except that the vocabulary size is now $2p$. It inspires us to apply Topic-SCORE to the stacked word count vectors $X_i = [(X_i^h)', (X_i^{ai})'] \in \mathbb{R}^{2p}$. Let $\widehat{A} \in \mathbb{R}^{2p \times K}$ be the estimated topic matrix. We take its first $p$ rows as $\widehat{A}^h$ and the last $p$ rows as $\widehat{A}^{ai}$. The remaining steps of obtaining word lists are the same as in (6).

**Remark 3**: We also define a simple classifier after obtaining $\widehat{S}$. Let $d \in \mathbb{R}^p$ be the empirical word frequency vector of a test document. Let $(\widehat{A}_{\widehat{S}}^h, \widehat{A}_{\widehat{S}}^{ai}, d_{\widehat{S}})$ be the counterpart of $(\widehat{A}^h, \widehat{A}^{ai}, d)$ restricted to the rows in $\widehat{S}$. Compute $\min_w \{\|\widehat{A}_{\widehat{S}}^h w - d_{\widehat{S}}\|\}$ and $\min_w \{\|\widehat{A}_{\widehat{S}}^{ai} w - d_{\widehat{S}}\|\}$, subject to that $w$ is nonnegative and $\mathbf{1}_K' w = 1$. We classify the test document to human or hwAI, depending on which of the above two quantities is smaller. We call it the ewTS (equal-weight Topic-SCORE) classifier, while naming the prompting approach that incorporates $\widehat{S}_1(t)$ and $\widehat{S}_2(t)$ the ewTS-LLM classifier.

## 3 NUMERICAL EXPERIMENTS

**Data sets, and generation of AI-edited and AI-written text.** We use three data sets. The first one is MADStat Ji et al. (2022); Ke et al. (2024) (link) which contains the title and text abstracts of

Table 1: Performance comparison with baselines. In each row, the method achieving the highest accuracy is bolded, and the method achieving the second highest accuracy is marked with ∗.

| Data set | Source LLM | HC | | HC-GPT | | DetectGPT | | Binoculars | | Detective | | MPU | | RoBERTa | |
|---|---|---|---|---|---|---|---|---|---|---|---|---|---|---|---|
| | | F1 | Accuracy | F1 | Accuracy | F1 | Accuracy | F1 | Accuracy | F1 | Accuracy | F1 | Accuracy | F1 | Accuracy |
| MADStat | GPT-4o-mini | 0.8264 | 0.8497 | 0.9251 | **0.9215** | 0.5411 | 0.5711 | 0.6698 | 0.5116 | 0.8612 | 0.8432 | 0.8853 | 0.8747* | 0.6667 | 0.5023 |
| | DeepSeek-V3 | 0.8761 | 0.8695 | 0.9050 | 0.9068 | 0.6084 | 0.6123 | 0.6643 | 0.5000 | 0.8312 | 0.8333 | 0.8674 | 0.8785* | 0.8057 | 0.7826 |
| | Claude Haiku | 0.5860 | 0.6970 | 0.9222 | **0.9207** | 0.6087 | 0.6128 | 0.6916 | 0.5581 | 0.8711 | 0.8723* | 0.8625 | 0.8413 | 0.8054 | 0.7807 |
| Movie | GPT-4o-mini | 0.8996 | 0.9011* | 0.9342 | **0.9363** | 0.6252 | 0.7034 | 0.7121 | 0.7162 | 0.9012 | 0.9002 | 0.7294 | 0.7877 | 0.7531 | 0.8022 |
| | DeepSeek-V3 | 0.9116 | 0.9054 | 0.9223 | 0.9214* | 0.3315 | 0.5792 | 0.6259 | 0.4608 | 0.8112 | 0.8219 | 0.8974 | 0.8868 | 0.9196 | **0.9269** |
| | Claude Haiku | 0.8118 | 0.8705 | 0.9227 | **0.9216** | 0.6498 | 0.6435 | 0.6434 | 0.5000 | 0.8912 | 0.8942* | 0.8624 | 0.8427 | 0.8569 | 0.8336 |
| Rewrite | Llama-2 | 0.8162 | 0.8275 | 0.9389 | **0.9380** | 0.6640 | 0.5970 | 0.6950 | 0.5690 | 0.8692 | 0.8630 | 0.7213 | 0.7328 | 0.8502 | 0.8701* |
| | Llama-3 | 0.6790 | 0.7505 | 0.9900 | **0.9900** | 0.6650 | 0.6030 | 0.7100 | 0.6020 | 0.9207 | 0.9220* | 0.7437 | 0.7528 | 0.8475 | 0.8675 |
| | GPT-3.5 | 0.9319 | 0.9300* | 0.9630 | **0.9630** | 0.0960 | 0.5080 | 0.6636 | 0.5010 | 0.8719 | 0.8754 | 0.5725 | 0.6132 | 0.8506 | 0.8698 |

83331 papers published in 36 statistics-related journals in between 1975 and 2015. We take a random sample of this dataset to include authors having at least 20 papers which gives us a final dataset of 2145 documents. The second is an Amazon movie review dataset (link). We mainly focus on the "text review" column containing the actual reviews as well as the "userID" column which uniquely identifies the reviewer. We again take a random sample of userID having more than 20 reviews and end up with a dataset of 3146 reviews. The third one is CUDRT (Tao et al., 2024) (link), a recent benchmark for AI text detection. It crafts different datasets for different tasks such as Translation, Rewrite, and so on. The Rewrite operation is the closest to our definition of hwAI, so we use three of their crafted Rewrite data sets, including BBC news and thesis from arXiv.

In the CUDRT-Rewrite data set, the AI-edited content was given. We used three of them, produced by Llama3, Llama2 and GPT-3.5, respectively. For the MADStat and Movie data sets, we generated the AI-edited content by ourself. Specifically, we prompted three different LLMs, Claude-Haiku, GPT-4o-mini and Deepseek-V3 with the prompt: *"Given the following abstract, make some revisions. Make sure not to change the length too much."* This gave a total of 9 data sets, as shown in Table 1.

The MADStat data set provides paper titles additional to abstracts. We leverage this to create purely AI-written text by providing the title to an LLM and asking it to write an abstract for this title. Now, we have both AI-edited and AI-written text for MADStat. We will mainly use MADStat to explore our own methods, while using all (MADStat, Movie, and Rewrite) for comparison with baselines.

**Comparison with baseline methods.** We consider several baselines for fair comparison. The first is DetectGPT (Mitchell et al., 2023). It uses a scoring model to compute log-probabilities and a mask-filling model to produce perturbations. We use *gpt2-medium* for scoring and *T5* for mask-filling, with algorithm parameters (e.g., number of perturbations) following the default values in the code. DetectGPT only outputs a statistic for each test document. To actually use it for classification, we need to choose a threshold. Some papers chose to report the AUROC for DetectGPT without setting an explicit threshold. However, since our methods directly output the classification decision, we must choose a threshold for DetectGPT to enable the comparison. We choose the threshold by minimizing the training error over a grid of 100 equally spaced thresholds. Therefore, even though DetectGPT is a zero-shot classifier, we have used the training data for threshold choice, giving more favor to this method. The other baselines include MPU (Tian et al.), RoBERTa, Binoculars (Hans et al., 2024), Ghostbuster (Verma et al., 2024), and Detective (Guo et al., 2024b). RoBERTa is a state-of-art classification method for AI-text detection. Following Kumarage et al. (2023), we use the RoBERTa-Stylo version and fine-tune it on training data following the standard procedure. MPU Tian et al. utilizes a special loss function to combine with RoBERTa. Binoculars is a zero-shot method using cross-perplexity between model pairs. Ghostbuster uses structured n-gram and probability features from multiple models. Detective employs multi-level contrastive learning for distinguishing AI-generated text from real data. For our own methods, we include HC and HC-GPT (see Section 2.1), both entirely tuning-free. The LLM used for classification is GPT-4o-mini (in all experiments below, we always use GPT-4o-mini as the default LLM to use, unless another LLM is mentioned). For MADStat and Movie, we sample 80% of documents for training and 20% for testing. For Rewrite, we use the same train-test split as in the original paper (Tao et al., 2024). The F1-score and accuracy are reported in Table 1. HC-GPT has the best accuracy in 8 out of 9 settings.

**Cross-domain robustness:** We also conducted cross-domain experiments to evaluate robustness to domain shift. We used all abstracts in MADStat for training and all movie reviews for testing. The performance comparison is shown in Table 2. While the performance of our method declines

when moving from in-domain to out-of-domain, it still significantly outperforms zero-shot detectors, demonstrating reasonable robustness to domain shift.

| Method | Accuracy | F1-score |
|---|---|---|
| HC-GPT (cross-domain) | 0.823 | 0.831 |
| Binoculars (cross-domain) | 0.710 | 0.701 |
| HC-GPT (in-domain) | 0.936 | 0.934 |
| Binoculars (in-domain) | 0.712 | 0.716 |

Table 2: Cross-domain performance: training on MADStat, testing on Movie reviews.

|  | Human vs. AI | | Human vs. HwAI | |
|---|---|---|---|---|
|  | HC | ewTS | HC | ewTS |
| 1 | (-) findings | (-) contributed | (-) additionally | (+) connect |
| 2 | (-) practical | (-) learned | (+) used | (-) additionally |
| 3 | (-) framework | (+) followed | (+) shown | (-) demonstrate |
| 4 | (-) various | (-) failed | (-) effectively | (+) conversely |
| 5 | (-) traditional | (-) statistical | (-) demonstrate | (+) derive |
| 6 | (-) techniques | (-) framework | (+) considered | (-) contain |
| 7 | (-) statistical | (-) findings | (-) introduce | (-) conducts |
| 8 | (-) novel | (+) hard | (-) utilizing | (-) characterized |
| 9 | (-) demonstrate | (-) distribution | (-) novel | (+) author |
| 10 | (-) comprehensive | (-) traditional | (-) scenarios | (-) findings |
| | . . . | . . . | . . . | . . . |
| | (135 in total) | (403 in total) | (69 in total) | (309 in total) |

Table 3: Most discriminative words in MADStat (+: human-indicative, -: AI-indicative).

**The selected words:** We study the selected word lists by HC and ewTS. We focused on MADStat and used the AI-edited (hwAI) and purely AI-written (AI) abstracts, both generated by GPT-4o-mini. We selected 15000 abstracts, which came from many different authors. Each abstract and its AI counterpart (the AI or hwAI version) form a pair. We randomly assigned 80% of document pairs for training and 20% for testing. Except ewTS and ewTS-GPT, the other methods don't use pairing information in training. For these two methods, we use a pair-reserving 5-fold cross-validation to select the number of topics $K$ and the threshold $t$. No pairing information is used in testing (same for other settings below). Table 3 shows the top ten words selected by HC and ewTS, respectively. For HC, words are ranked in the increasing order of p-values; and for ewTS, they are ranked in the descending order of $T_j$. We find that words like 'findings', 'framework' and 'demonstrate' are favored by AI. In comparison, there are fewer human-indicative words selected. One reason is that the human-written abstracts come from diversified authors and don't have a strong consistent pattern.

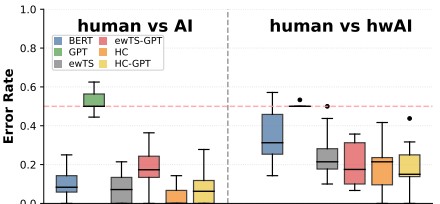

| Train | Test | GPT | HC-GPT |
|---|---|---|---|
| James O. Ramsay | Stephen Fienberg | 0.375 | 0.344 |
| Peter Bickel | Stephen Feinberg | 0.375 | 0.172 |
| Atanu Biswas | Stephen Feinberg | 0.375 | 0.197 |
| Stephen Fienberg | Wolfgang Härdle | 0.340 | 0.264 |
| Stephen Fienberg | Robert Serfling | 0.321 | 0.089 |
| Stephen Fienberg | Jon Wellner | 0.368 | 0.226 |

Figure 4: The testing errors in the CAD setting. The boxplots are from results of 153 author pairs.

Table 4: The testing errors in the CAD setting (human-vs-hwAI) for some author pairs.

**The cross-author design (CAD):** We consider a setting where the training data have a small size and are heterogeneous from testing data. We still focus on MADStat, as it contains author information. We randomly selected $N = 18$ authors and considered the $\binom{N}{2} = 153$ author pairs. For each pair, we trained classifiers using all abstracts written by the first author, along with the AI counterparts; and the testing error were evaluated using all abstracts and AI counterparts of the second author. Since there are 153 errors for each method, we present the results with a boxplot per method; see Figure 4. Besides our four methods, we have also included two simple baselines. The first is a transfer-learning approach by training a classifier using BERT features. Specifically, we use the 'all-MiniLM-L6-v2' variant of a pre-trained BERT (Wang et al., 2020) to get 384-dimensional features for each document. We then build a classifier on these features, using a Random Forest with 100 trees. The second is directly prompting GPT-4o-mini without any word list inserted. The human-vs-AI setting uses the AI-written abstracts with given titles, and human-vs-hwAI is the same as before. For human-vs-AI, HC has a remarkable performance, even better than HC-GPT. One possible reason is that HC is most sensitive to author writing styles. In the current CAD setting, classifiers often incur with unseen writing styles in testing documents, and HC is powerful in harnessing such signals. For human-vs-hwAI, HC-GPT is the best, and ewTS-GPT is the second best (in terms of median error). The direct prompting approach by GPT performs poorly. The transfer learning approach by BERT performs well for human-vs-AI but unsatisfactorily for human-vs-hwAI. Table 4 shows the errors for some author pairs. We don't observe clear patterns related to authors' research interests. The errors are mostly affected by author writing styles.

**The same-author design (SAD):** In this setting, we still focus on MADStat but let the training and testing documents come from the same author. We randomly selected $N = 15$ authors that have more than 30 abstracts in MADStat. For each author, we randomly assigned 80% of his/her abstracts (along with AI counterparts) to the training set and used the remaining 20% (along with AI counterparts) for testing. We restricted to authors with more than 30 abstracts to ensure that there are enough training documents (as only 80% of each author's abstracts are used for training). We consider using two different LLMs for generating the AI-content, GPT-4o-mini and Claude-Haiku. The LLM used in classification is always GPT-4o-mini. For each author, we obtain a testing error for each method. The boxplots based on 15 errors are shown in Figure 5. Most methods have similar performance on GPT-generated and Claude-generated content. Focusing on GPT-generated content, BERT performs well in the human-vs-AI problem setting, but much worse in human-vs-hwAI. The direct prompting approach is always the worst. For human-vs-AI, HC is the best. For human-vs-hwAI, ewTS-GPT and HC-GPT have the smallest median errors.

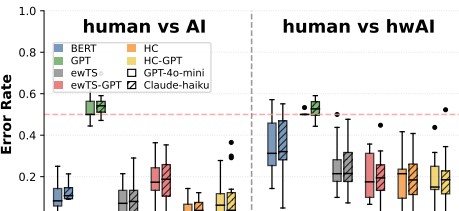

Figure 5: The testing errors in the SAD setting using both GPT-4o-mini and Claude-Haiku to generate the AI content. Boxplots are based on the results on 15 authors.

| LLM | Without HC | With HC |
|---|---|---|
| Claude-Haiku | 0.071 (0.0178) | 0.017 (0.0172) |
| DeepSeek-R1 | 0.364 (0.013) | 0.124 (0.021) |
| Gemini-1.5-flash | 0.206 (0.019) | 0.089 (0.029) |
| GPT-4o-mini | 0.513 (0.0163) | 0.098 (0.026) |
| LLaMA-8B | 0.599 (0.028) | 0.402 (0.018) |

Table 5: The results about using different LLMs in classification (but the AI-content is still generated by GPT-4o-mini). For each LLM, we compare the direct prompting ('Without HC') and our word-list-assisted prompting ('With HC').

**Comparison of 5 LLMs in classification:** In previous experiments, we prompt GPT-4o-mini for classification. In this experiment, we consider using other LLMs (see Table 5). For each LLM, we consider both prompting without/with the HC-selected word list. The training/testing data are the same as those in the previous SAD experiment, with the AI- and hwAI- content still generated by GPT-4o-mini. Table 5 shows that across all LLMs tested, adding a discriminative word list from HC consistently improves performance. Claude and Gemini already perform reasonably well with direct prompting, but still benefit from HC-based guidance. For weaker models like GPT and LLaMA, the improvements are dramatic, e.g., for GPT, the error drops from $0.513$ to $0.098$. This supports our main claim: even when the LLM alone struggles, guiding it with statistically selected features can greatly enhance accuracy with minimal additional cost.

**LLM-pairs for editing and classification:** We consider using LLM1 to generate AI-content and LLM2 for classification. In our previous experiments, we either fix LLM2 and vary LLM1 (Table 1 and Figure 5) or fix LLM1 and vary LLM2 (Table 5). In this experiment, we let both LLMs range in ChatGPT, DeepSeek-V3, and Claude-Haiku, giving $3 \times 3 = 9$ combinations. The results are in Table 6. Without HC-guided prompting (right half of Table 6), there is no clear evidence that a given LLM is particularly effective at detecting the outputs it generated—performance is low and inconsistent. For example, ChatGPT achieves only $54.5\%$ accuracy and $16.7\%$ F1 when trying to detect its own generations, and DeepSeek fails entirely with $50\%$ accuracy and zero F1 across all cases. In contrast, with HC-assisted prompting (left half), we observe a strong diagonal pattern: each LLM achieves its best performance when classifying documents generated by itself (highlighted in bold). This suggests that while direct prompting is unreliable, adding HC-selected discriminative word lists enables each model to more effectively recognize its own generation style.

Table 6: Cross-LLM comparison (row: classification LLM, column: source LLM, each entry is accuracy / F1). For each column, the best method is in marked in bold.

| | ChatGPT | Claude | DeepSeek | | ChatGPT | Claude | DeepSeek |
|---|---|---|---|---|---|---|---|
| HC-ChatGPT | **.932 / .930** | .909 / .905 | .886 / .878 | ChatGPT | .545 / .167 | .682 / .533 | .523 / .087 |
| HC-Claude | .932 / .927 | **.932 / .927** | .795 / .743 | Claude | **.659 / .483** | **.795 / .743** | **.614 / .370** |
| HC-DeepSeek | .864 / .857 | .795 / .769 | **.932 / .933** | DeepSeek | .500 / .000 | .500 / .000 | .500 / .000 |

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
