# Supplement for the article "When Machines Write: A method for detecting AI-edited text"

## A    GitHub Link for Code

The scripts for all methods used in the main paper are available in the anonymized GitHub repository `https://anonymous.4open.science/r/AIvsHuman-3B67/`.

All data sets in our numerical experiments are publicly available. The links for downloading these data sets have been included in the main article.

## B    Additional Results for Table 1

Table 1 of the main article presents a comparison of HC and HC-GPT with three baseline methods, DetectGPT, RoBERTa, and MPU. The results of ewTS and ewTS-GPT are omitted in the main article due to space limit. They are presented here in Table B.1. These methods require choosing two tuning parameters, the number of topics $K$, and the threshold $t$. We always choose them by cross validation, with the grid of $K \in \{3, 4, 5, 6\}$ and $t \in \{0.001, 0.005, 0.01, 0.1\}$.

We recall that HC-GPT and ewTS-GPT are the methods we recommend. In comparison, HC-GPT has a better performance than ewTS-GPT in this experiment. One reason is that ewTS-GPT requires choosing tuning parameters by cross-validation, while HC-GPT is completely tuning-free. When the training sample is larger or from different domains, ewTS-GPT may perform better.

Table B.1: Performance of ewTS and ewTS-GPT for the experiment in Table 1 in the main paper.

| Data set | Source LLM | ewTS | | ewTS-GPT | |
|---|---|---|---|---|---|
| | | F1 | Accuracy | F1 | Accuracy |
| MADStat | GPT-4o-mini | 0.8133 | 0.8163 | 0.8434 | 0.8623 |
| | DeepSeek-V3 | 0.8137 | 0.7977 | 0.8812 | 0.8770 |
| | Claude Haiku | 0.7719 | 0.7651 | 0.7812 | 0.7780 |
| Movie | GPT-4o-mini | 0.6546 | 0.6612 | 0.7710 | 0.7872 |
| | DeepSeek-V3 | 0.6821 | 0.6710 | 0.7912 | 0.8021 |
| | Claude Haiku | 0.6544 | 0.6555 | 0.7301 | 0.7311 |
| Rewrite | Llama-2 | 0.8024 | 0.8205 | 0.8712 | 0.8801 |
| | Llama-3 | 0.8940 | 0.9000 | 0.9101 | 0.9012 |
| | GPT-3.5 | 0.9020 | 0.8955 | 0.9321 | 0.9212 |

We present in Table B.2 the average accuracy for each method by combining all results in different data sets (e.g., the MADStat accuracy ewTS-GPT is the average of Rows 3-5 of Column 4 in Table B.1, and the average accuracy of ewTS-GPT is the average over Rows 3-11 of Column 4 in Table B.1). We observe that HC-GPT performs the best in each data set. Which method is the second best varies with data set. RoBERTa, MPU, and ewTS-GPT are the second best in MADStat, Movie, and Rewrite, respectively. In terms of the average accuracy over all data sets, ewTS-GPT is the second best.

## C    Full Selected Word Lists for Table 3

Table 3 of the main article presents some selected words by HC and ewTS on the MADStat data set (consisting of academic abstracts). We present the full word lists selected by HC in Table C.3. The

Table B.2: The per-data-set average accuracy for different methods.

| Data set | HC-GPT | ewTS-GPT | DetectGPT | RoBERTa | MPU |
|----------|--------|----------|-----------|---------|-----|
| MADStat | 0.916 | 0.839 | 0.599 | 0.865 | 0.689 |
| Movie | 0.926 | 0.773 | 0.642 | 0.839 | 0.854 |
| Rewrite | 0.964 | 0.901 | 0.569 | 0.700 | 0.869 |
| Average | 0.935 | 0.838 | 0.603 | 0.801 | 0.804 |

full word lists selected by ewTS are too long (309 words for human-vs-hwAI, and 403 words for human-vs-AI), hence not presented here.

Table C.3: The selected word lists by HC. They are ranked in the same way as described in Table 3. Words that are underscored are the human-indicative words, with the remaining AI-indicative words.

| Setting | Word List |
|---------|-----------|
| human-vs-hwAI | (69 words) *additionally, demonstrate, findings, used, scenarios, utilizing, introduce, shown, explore, consider, significant, specific, effectively, notably, use, specifically, exhibit, given, establish, considered, demonstrates, challenges, various, obtained, obtain, address, framework, straightforward, novel, derive, employing, particularly, problem, utilized, useful, utilize, focus, present, enhance, presents, addresses, significantly, introduces, simple, effectiveness, studied, examines, explores, regarding, challenge, insights, indicate, crucial, exhibits, important, demonstrating, particular, valuable, numerous, realworld, addition, way, alongside, focusing, furthermore, offers, discussed, examine, aimed* |
| human-vs-AI | (135 words) *findings, framework, paper, realworld, traditional, demonstrate, statistical, practical, various, techniques, novel, comprehensive, implications, insights, applications, understanding, fields, scenarios, theoretical, research, robustness, highlighting including, extensive, simulations, providing, contributes, robust, accuracy, practitioners, enhance, analysis, researchers, complex, datasets, additionally, particularly, modeling, provide, demonstrating, significant, performance, c, challenges, effectiveness, offering, studies, methodologies, approach, elsevier, enhancing, presents, explore, tool, contribute, methodology, underlying, advanced, work, structures, potential, leverages, future, derive, estimation, utilizing, highlight, efficacy, results, valuable, introduce, underscore, crucial, methods, applicability, decisionmaking, inference, enhances, allowing, leveraging, illustrate, settings, reliable, introduces, contexts, employing, significantly, finance, approaches, diverse, indicate, inherent, investigates, ultimately, integrates, reveal, given, behavior, interpretability, paving, used, presence, properties, relationships, strategies, utility, context, importance, addressing, illustrating, article, tools, reliability, obtained, simulation, existing, effectively, tailored, focusing, apply, establish, example, problem, broader, complexities, incorporating, frameworks, realm, field, general, showcasing, incorporates, rigorous, characteristics, dealing* |

From Table C.3 we observe that HC selects more words in the human-vs-AI setting, most of which are AI-indicative words. HC does recrui more human-indicative words in the human-vs-hwAI setting.

## D COMPLETE RESULTS FOR TABLE 4

Table 4 of the main article presents the testing errors for some author pairs in the Cross-Author Design (CAD). We now display the complete results for all author pairs in Table D.4.

Table D.4: Pair-wise results for the CAD experiment. The first column is the author whose abstracts are used for training, and the second column is the author for testing. In each row, the bolded number indicated the highest accuracy achieved by one of the four methods.

| Author 1 | Author 2 | Train Size | Test Size | ewTS | ewTS-GPT | HC | HC-GPT |
|----------|----------|------------|-----------|------|----------|-----|--------|
| James O. Ramsay | Peter Bickel | 30 | 72 | 0.569 | 0.590 | 0.639 | **0.688** |
| James O. Ramsay | Atanu Biswas | 30 | 38 | 0.632 | 0.658 | 0.579 | **0.868** |

| | | | | | | | |
|---|---|---|---|---|---|---|---|
| James O. Ramsay | Stephen Fienberg | 30 | 32 | 0.625 | 0.641 | 0.625 | **0.656** |
| James O. Ramsay | Yanqing Sun | 30 | 30 | 0.617 | 0.717 | 0.750 | **0.783** |
| James O. Ramsay | Wolfgang Härdle | 30 | 53 | 0.547 | 0.632 | 0.623 | **0.764** |
| James O. Ramsay | Nicholas P. Jewell | 30 | 42 | 0.583 | 0.738 | **0.750** | 0.655 |
| James O. Ramsay | Manlai Tang | 30 | 36 | 0.569 | 0.625 | 0.583 | **0.806** |
| James O. Ramsay | Hansheng Wang | 30 | 26 | 0.654 | 0.769 | 0.577 | **0.865** |
| James O. Ramsay | Robert Serfling | 30 | 28 | **0.696** | 0.643 | 0.661 | **0.696** |
| James O. Ramsay | Zakkula Govindarajulu | 30 | 35 | 0.700 | 0.586 | 0.600 | **0.786** |
| James O. Ramsay | Johan Segers | 30 | 32 | 0.531 | 0.578 | 0.641 | **0.766** |
| James O. Ramsay | Philippe Vieu | 30 | 39 | 0.641 | 0.667 | 0.603 | **0.795** |
| James O. Ramsay | Michel Talagrand | 30 | 55 | 0.623 | 0.536 | 0.673 | **0.727** |
| James O. Ramsay | Jon Wellner | 30 | 53 | 0.604 | 0.613 | 0.632 | **0.698** |
| James O. Ramsay | Yufeng Liu | 30 | 35 | 0.714 | 0.614 | 0.729 | **0.814** |
| James O. Ramsay | Yongtao 1 Guan | 30 | 35 | 0.443 | 0.614 | 0.700 | **0.758** |
| James O. Ramsay | Qi Man Shao | 30 | 52 | 0.567 | 0.549 | 0.625 | **0.731** |
| Peter Bickel | Atanu Biswas | 72 | 38 | 0.789 | 0.789 | 0.579 | **0.895** |
| Peter Bickel | Stephen Fienberg | 72 | 32 | 0.719 | 0.813 | 0.625 | **0.828** |
| Peter Bickel | Yanqing Sun | 72 | 30 | 0.750 | **0.817** | 0.733 | 0.800 |
| Peter Bickel | Wolfgang Härdle | 72 | 53 | 0.736 | 0.792 | 0.613 | **0.868** |
| Peter Bickel | Nicholas P. Jewell | 72 | 42 | 0.738 | 0.810 | 0.738 | **0.845** |
| Peter Bickel | Manlai Tang | 72 | 36 | 0.694 | **0.861** | 0.597 | 0.778 |
| Peter Bickel | Hansheng Wang | 72 | 26 | 0.692 | **0.865** | 0.654 | **0.865** |
| Peter Bickel | Robert Serfling | 72 | 28 | 0.857 | **0.911** | 0.661 | **0.911** |
| Peter Bickel | Zakkula Govindarajulu | 72 | 35 | 0.771 | 0.786 | 0.614 | **0.943** |
| Peter Bickel | Johan Segers | 72 | 32 | 0.719 | **0.813** | 0.625 | 0.797 |
| Peter Bickel | Philippe Vieu | 72 | 39 | 0.821 | **0.885** | 0.603 | **0.885** |
| Peter Bickel | Michel Talagrand | 72 | 55 | 0.723 | 0.827 | 0.627 | **0.855** |
| Peter Bickel | Jon Wellner | 72 | 53 | 0.755 | **0.868** | 0.642 | **0.868** |
| Peter Bickel | Yufeng Liu | 72 | 35 | 0.843 | 0.829 | 0.686 | **0.857** |
| Peter Bickel | Yongtao 1 Guan | 72 | 35 | 0.786 | **0.929** | 0.700 | 0.786 |
| Peter Bickel | Qi Man Shao | 72 | 52 | 0.692 | 0.769 | 0.606 | **0.846** |
| Atanu Biswas | Stephen Fienberg | 38 | 32 | 0.578 | 0.641 | 0.625 | **0.813** |
| Atanu Biswas | Yanqing Sun | 38 | 30 | 0.700 | 0.717 | 0.750 | **0.800** |
| Atanu Biswas | Wolfgang Härdle | 38 | 53 | 0.632 | 0.679 | 0.632 | **0.783** |
| Atanu Biswas | Nicholas P. Jewell | 38 | 42 | 0.571 | 0.667 | **0.738** | 0.738 |
| Atanu Biswas | Manlai Tang | 38 | 36 | 0.639 | 0.639 | 0.583 | **0.708** |
| Atanu Biswas | Hansheng Wang | 38 | 26 | 0.673 | **0.731** | 0.635 | 0.692 |
| Atanu Biswas | Robert Serfling | 38 | 28 | 0.768 | 0.696 | 0.661 | **0.786** |
| Atanu Biswas | Zakkula Govindarajulu | 38 | 35 | 0.686 | 0.700 | 0.614 | **0.829** |
| Atanu Biswas | Johan Segers | 38 | 32 | 0.594 | 0.609 | 0.609 | **0.766** |
| Atanu Biswas | Philippe Vieu | 38 | 39 | 0.731 | 0.718 | 0.615 | **0.808** |
| Atanu Biswas | Michel Talagrand | 38 | 55 | 0.609 | 0.600 | 0.618 | **0.801** |
| Atanu Biswas | Jon Wellner | 38 | 53 | 0.575 | 0.670 | 0.594 | **0.755** |
| Atanu Biswas | Yufeng Liu | 38 | 35 | 0.529 | 0.629 | **0.729** | 0.643 |
| Atanu Biswas | Yongtao 1 Guan | 38 | 35 | 0.600 | 0.614 | 0.743 | **0.800** |
| Atanu Biswas | Qi Man Shao | 38 | 52 | 0.654 | 0.721 | 0.577 | **0.798** |
| Stephen Fienberg | Yanqing Sun | 32 | 30 | 0.650 | 0.700 | 0.733 | **0.800** |
| Stephen Fienberg | Wolfgang Härdle | 32 | 53 | 0.745 | **0.764** | 0.660 | 0.736 |
| Stephen Fienberg | Nicholas P. Jewell | 32 | 42 | 0.750 | **0.786** | 0.774 | 0.738 |
| Stephen Fienberg | Manlai Tang | 32 | 36 | 0.722 | 0.722 | 0.583 | **0.750** |
| Stephen Fienberg | Hansheng Wang | 32 | 26 | 0.654 | 0.750 | 0.654 | **0.769** |
| Stephen Fienberg | Robert Serfling | 32 | 28 | 0.732 | 0.750 | 0.679 | **0.911** |
| Stephen Fienberg | Zakkula Govindarajulu | 32 | 35 | 0.800 | 0.786 | 0.629 | **0.929** |
| Stephen Fienberg | Johan Segers | 32 | 32 | 0.703 | **0.797** | 0.625 | 0.766 |
| Stephen Fienberg | Philippe Vieu | 32 | 39 | 0.782 | **0.833** | 0.615 | **0.833** |
| Stephen Fienberg | Michel Talagrand | 32 | 55 | 0.691 | 0.782 | 0.601 | **0.827** |
| Stephen Fienberg | Jon Wellner | 32 | 53 | 0.755 | 0.736 | 0.632 | **0.774** |
| Stephen Fienberg | Yufeng Liu | 32 | 35 | 0.686 | 0.771 | **0.800** | 0.800 |
| Stephen Fienberg | Yongtao 1 Guan | 32 | 35 | 0.500 | 0.657 | 0.700 | **0.729** |
| Stephen Fienberg | Qi Man Shao | 32 | 52 | 0.702 | 0.750 | 0.606 | **0.798** |
| Yanqing Sun | Wolfgang Härdle | 30 | 53 | 0.623 | 0.764 | 0.613 | **0.774** |
| Yanqing Sun | Nicholas P. Jewell | 30 | 42 | 0.619 | 0.667 | **0.750** | 0.738 |

| | | | | | | | |
|---|---|---|---|---|---|---|---|
| Yanqing Sun | Manlai Tang | 30 | 36 | 0.625 | **0.653** | 0.569 | 0.611 |
| Yanqing Sun | Hansheng Wang | 30 | 26 | 0.673 | **0.712** | 0.673 | 0.558 |
| Yanqing Sun | Robert Serfling | 30 | 28 | 0.625 | **0.768** | 0.661 | **0.768** |
| Yanqing Sun | Zakkula Govindarajulu | 30 | 35 | 0.657 | 0.786 | 0.586 | **0.829** |
| Yanqing Sun | Johan Segers | 30 | 32 | 0.500 | **0.766** | 0.609 | 0.750 |
| Yanqing Sun | Philippe Vieu | 30 | 39 | 0.718 | 0.769 | 0.603 | **0.808** |
| Yanqing Sun | Michel Talagrand | 30 | 55 | 0.500 | 0.782 | 0.673 | **0.873** |
| Yanqing Sun | Jon Wellner | 30 | 53 | 0.632 | 0.745 | 0.632 | **0.830** |
| Yanqing Sun | Yufeng Liu | 30 | 35 | 0.700 | 0.571 | **0.757** | 0.514 |
| Yanqing Sun | Yongtao 1 Guan | 30 | 35 | 0.657 | 0.571 | **0.729** | 0.557 |
| Yanqing Sun | Qi Man Shao | 30 | 52 | 0.673 | 0.808 | 0.577 | **0.827** |
| Wolfgang Härdle | Nicholas P. Jewell | 53 | 42 | 0.679 | **0.821** | 0.750 | **0.821** |
| Wolfgang Härdle | Manlai Tang | 53 | 36 | 0.736 | **0.778** | 0.625 | 0.708 |
| Wolfgang Härdle | Hansheng Wang | 53 | 26 | 0.712 | **0.885** | 0.635 | 0.827 |
| Wolfgang Härdle | Robert Serfling | 53 | 28 | 0.661 | **0.893** | 0.625 | 0.875 |
| Wolfgang Härdle | Zakkula Govindarajulu | 53 | 35 | 0.700 | 0.829 | 0.643 | **0.886** |
| Wolfgang Härdle | Johan Segers | 53 | 32 | 0.641 | **0.828** | 0.609 | 0.766 |
| Wolfgang Härdle | Philippe Vieu | 53 | 39 | 0.795 | **0.872** | 0.603 | 0.795 |
| Wolfgang Härdle | Michel Talagrand | 53 | 55 | 0.673 | 0.809 | 0.682 | **0.855** |
| Wolfgang Härdle | Jon Wellner | 53 | 53 | 0.736 | 0.858 | 0.613 | **0.906** |
| Wolfgang Härdle | Yufeng Liu | 53 | 35 | 0.729 | **0.771** | 0.743 | 0.657 |
| Wolfgang Härdle | Yongtao 1 Guan | 53 | 35 | 0.714 | 0.714 | 0.686 | **0.771** |
| Wolfgang Härdle | Qi Man Shao | 53 | 52 | 0.692 | **0.827** | 0.644 | **0.827** |
| Nicholas P. Jewell | Manlai Tang | 42 | 36 | 0.583 | **0.847** | 0.611 | 0.764 |
| Nicholas P. Jewell | Hansheng Wang | 42 | 26 | 0.769 | **0.788** | 0.654 | **0.788** |
| Nicholas P. Jewell | Robert Serfling | 42 | 28 | 0.625 | 0.839 | 0.607 | **0.929** |
| Nicholas P. Jewell | Zakkula Govindarajulu | 42 | 35 | 0.743 | 0.800 | 0.629 | **0.957** |
| Nicholas P. Jewell | Johan Segers | 42 | 32 | 0.656 | 0.797 | 0.625 | **0.828** |
| Nicholas P. Jewell | Philippe Vieu | 42 | 39 | 0.744 | **0.872** | 0.577 | 0.833 |
| Nicholas P. Jewell | Michel Talagrand | 42 | 55 | 0.718 | 0.727 | 0.645 | **0.809** |
| Nicholas P. Jewell | Jon Wellner | 42 | 53 | 0.689 | 0.849 | 0.623 | **0.943** |
| Nicholas P. Jewell | Yufeng Liu | 42 | 35 | 0.657 | **0.771** | 0.757 | 0.686 |
| Nicholas P. Jewell | Yongtao 1 Guan | 42 | 35 | 0.629 | 0.729 | 0.729 | **0.786** |
| Nicholas P. Jewell | Qi Man Shao | 42 | 52 | 0.683 | 0.712 | 0.625 | **0.875** |
| Manlai Tang | Hansheng Wang | 36 | 26 | 0.769 | **0.865** | 0.673 | **0.865** |
| Manlai Tang | Robert Serfling | 36 | 28 | 0.643 | 0.661 | 0.661 | **0.732** |
| Manlai Tang | Zakkula Govindarajulu | 36 | 35 | 0.686 | 0.729 | 0.600 | **0.814** |
| Manlai Tang | Johan Segers | 36 | 32 | 0.688 | **0.781** | 0.625 | 0.719 |
| Manlai Tang | Philippe Vieu | 36 | 39 | 0.718 | 0.756 | 0.590 | **0.769** |
| Manlai Tang | Michel Talagrand | 36 | 55 | **0.709** | **0.709** | 0.618 | 0.700 |
| Manlai Tang | Jon Wellner | 36 | 53 | 0.651 | **0.792** | 0.613 | 0.774 |
| Manlai Tang | Yufeng Liu | 36 | 35 | 0.829 | 0.786 | 0.771 | **0.886** |
| Manlai Tang | Yongtao 1 Guan | 36 | 35 | **0.714** | 0.629 | 0.671 | 0.657 |
| Manlai Tang | Qi Man Shao | 36 | 52 | 0.673 | 0.721 | 0.596 | **0.769** |
| Hansheng Wang | Robert Serfling | 26 | 28 | 0.643 | 0.625 | **0.661** | **0.661** |
| Hansheng Wang | Zakkula Govindarajulu | 26 | 35 | 0.714 | 0.671 | 0.600 | **0.829** |
| Hansheng Wang | Johan Segers | 26 | 32 | 0.688 | 0.719 | 0.641 | **0.750** |
| Hansheng Wang | Philippe Vieu | 26 | 39 | 0.705 | 0.692 | 0.603 | **0.756** |
| Hansheng Wang | Michel Talagrand | 26 | 55 | 0.782 | 0.564 | 0.682 | **0.791** |
| Hansheng Wang | Jon Wellner | 26 | 53 | 0.632 | 0.660 | 0.632 | **0.726** |
| Hansheng Wang | Yufeng Liu | 26 | 35 | 0.771 | 0.843 | 0.729 | **0.957** |
| Hansheng Wang | Yongtao 1 Guan | 26 | 35 | 0.686 | 0.729 | 0.700 | **0.771** |
| Hansheng Wang | Qi Man Shao | 26 | 52 | 0.702 | 0.692 | 0.596 | **0.769** |
| Robert Serfling | Zakkula Govindarajulu | 28 | 35 | 0.557 | 0.843 | 0.600 | **0.943** |
| Robert Serfling | Johan Segers | 28 | 32 | 0.609 | 0.812 | 0.656 | **0.859** |
| Robert Serfling | Philippe Vieu | 28 | 39 | 0.718 | 0.795 | 0.615 | **0.872** |
| Robert Serfling | Michel Talagrand | 28 | 55 | 0.691 | 0.755 | 0.664 | **0.873** |
| Robert Serfling | Jon Wellner | 28 | 53 | 0.679 | 0.792 | 0.594 | **0.821** |
| Robert Serfling | Yufeng Liu | 28 | 35 | 0.714 | 0.757 | 0.743 | **0.771** |
| Robert Serfling | Yongtao 1 Guan | 28 | 35 | 0.671 | 0.743 | 0.714 | **0.857** |

| | | | | | | | |
|---|---|---|---|---|---|---|---|
| Robert Serfling | Qi Man Shao | 28 | 52 | 0.663 | 0.808 | 0.596 | **0.856** |
| Zakkula Govindarajulu | Johan Segers | 35 | 32 | 0.766 | 0.781 | 0.641 | **0.797** |
| Zakkula Govindarajulu | Philippe Vieu | 35 | 39 | 0.718 | 0.821 | 0.615 | **0.885** |
| Zakkula Govindarajulu | Michel Talagrand | 35 | 55 | 0.700 | **0.836** | 0.682 | 0.827 |
| Zakkula Govindarajulu | Jon Wellner | 35 | 53 | 0.745 | 0.811 | 0.594 | **0.896** |
| Zakkula Govindarajulu | Yufeng Liu | 35 | 35 | **0.743** | 0.686 | 0.714 | 0.671 |
| Zakkula Govindarajulu | Yongtao 1 Guan | 35 | 35 | 0.600 | 0.600 | **0.686** | 0.643 |
| Zakkula Govindarajulu | Qi Man Shao | 35 | 52 | 0.721 | **0.885** | 0.577 | 0.846 |
| Johan Segers | Philippe Vieu | 32 | 39 | 0.705 | **0.795** | 0.615 | **0.795** |
| Johan Segers | Michel Talagrand | 32 | 55 | 0.782 | 0.782 | 0.682 | **0.800** |
| Johan Segers | Jon Wellner | 32 | 53 | 0.670 | **0.764** | 0.623 | 0.755 |
| Johan Segers | Yufeng Liu | 32 | 35 | 0.700 | 0.729 | 0.500 | **0.757** |