# OpenReview forum: "When Machines Write: A method for detecting AI-edited text"
_ICLR.cc/2026/Conference — ICLR 2026 Conference Withdrawn Submission_

### Official Review · Reviewer_o5dF · 2025-10-24

**Soundness:** 2
**Presentation:** 2
**Contribution:** 2
**Rating:** 2
**Confidence:** 4

**Summary:**

This paper introduces a prompt-based method for detecting AI-edited texts. It first selects word list critical to detection from the training set and prompt the LLM with this additional information for detection.

**Strengths:**

1. The task of detecting AI-edited texts is challenging.

2. The proposed method is relatively efficient compared with methods that require extensive training.

**Weaknesses:**

1. My concern about this paper is its assumption that most editing by AI is word replacement. I believe how AI edits texts depends on the instructions instead of its inner mechanism. I find the prompt for editing used in this work is "Given the following abstract, make some revisions. Make sure not to change the length too much." (line 348-349). I think "make some revisions" is not a normal instruction for editing in real life. Therefore, I question if the proposed method would generalize to different editing instructions that are practical in real-world applications.

2. The definition of "AI-edited text" is not clear. For example, if the text contains three paragraphs and you only ask the AI to edit one of them, would you call the whole text "AI-edited"? I suppose the core problem in such a question is to conduct fine-grained detection instead of giving an overall judgement about the whole text.

3. As a safety-related work, more analysis about the robustness of the proposed method should be condecuted.

**Questions:**

See the weakness section.

---

### Official Review · Reviewer_8EF1 · 2025-10-27

**Soundness:** 2
**Presentation:** 2
**Contribution:** 2
**Rating:** 4
**Confidence:** 3

**Summary:**

This paper tackles a more challenging detection problem, distinguishing between purely human-written text and human-written but AI-edited text (called as hwAI text), where conventional AI detectors often fail due to weaker signals. The authors propose a word-list-assisted prompting approach that enhances prompt-based detection using carefully selected discriminative words. They first observe that simple word-count features remain surprisingly effective for identifying hwAI text, and that prompting can achieve strong results when augmented with such word lists. To construct these lists, they introduce two feature selection methods grounded in large-scale multiple testing and topic modeling. Experiments across diverse domains demonstrate that the proposed method achieves strong and consistent performance in detecting AI-edited texts.

**Strengths:**

- Propose a new detection approach via a simple prompting with an optimized word list to discern human-written and human-edited-by-AI texts
- Thorough evaluations across multiple detectors and recent generators

**Weaknesses:**

- **Lack of justification for prompting approach:** The paper claims that prompt-based methods consistently outperform simple linear classifiers with the curated word list (L131), yet no numerical comparison is provided in the main results (Table 1). Without such evidence, the necessity of using prompts over linear classifiers remains unclear.
- **Missing evaluation on core detection task:** While distinguishing human vs. human-edited-by-AI texts is valuable, it is not shown whether the method can still detect human vs. machine-generated texts, which is the original and simpler task. The absence of this verification raises concerns about effectiveness of the proposed method.
- **Limited robustness analysis:** Despite claiming that discrete word lists prevent overfitting (L084), the work does not evaluate robustness against paraphrasing or adversarial perturbations, where such discrete cues are likely fragile.
- **Limited cross-(domain | model) detection evaluation:** Cross-domain results are presented but with only one baseline (Binoculars, zero-shot detector), making its robustness questionable. Likewise, there is no cross-model evaluation (e.g., testing on GPT-4o-mini edits with word lists from Gemini-1.5-Flash edits), which is essential given that the editing model is the key factor in this task.
- **Too simple edit prompt setting:** The paper observes that AI edits are mostly word replacements (L085), but this may depend heavily on the edit instruction. Indeed, the prompt in L348 (“Given the following abstract, make some revisions. Make sure not to change the length too much.”) is overly simple and does not reflect real-world editing intents such as improving readability or persuasiveness. This raises the questionable effectiveness of the proposed method in real-world scenarios, where people use more diverse prompting for AI edits.
- **Missing reference to relevant prior work:** The study does not refer to a prior work [1] first demonstrating that prompting LLMs can be an effective detection via in-context learning with retrieved similar texts, which should be discussed for a fair contextualization of the proposed approach.

---

References:

[1] Koike et al. OUTFOX: LLM-Generated Essay Detection Through In-Context Learning with Adversarially Generated Examples. AAAI 2024.

**Questions:**

See the weaknesses part

---

### Official Review · Reviewer_pbLK · 2025-11-01

**Soundness:** 2
**Presentation:** 1
**Contribution:** 2
**Rating:** 2
**Confidence:** 4

**Summary:**

The paper tackles detecting human-written, AI-edited text (hwAI) -- a harder setting than pure AI generation because signals are weaker. It proposes word-list-assisted prompting: select a sparse, signed vocabulary via multiple-testing (Higher Criticism) and topic-modeling, then include that list in the LLM prompt for hwAI detection.

**Strengths:**

**Mathematically principled feature selection:** They model per-word counts as Poisson, compute z-scores and p-values, then pick a tuning-free threshold via Higher Criticism -- with a p-value justification -- and sign words by the z-score, yielding an interpretable lexicon for rare/weak signals.

**Weaknesses:**

1. Weak Contribution: The authors evaluate detection only for HW-text vs. hwAI-text, not AI-text vs. AIhw-text. Moreover, prior work has already highlighted this problem.

2. Relevant Works Missing: There are existing studies on detecting AI-rewritten/AI-polished texts (e.g., [1, 2, 3]). The authors should clarify their positioning relative to these and evaluate on benchmarks like APT-Eval [1] and MixSet [2], and include RoBERTa-HPPT [3] as a baseline.

3. Strong Assumption: Authors assume edits are largely word replacements and count “natural words” without stemming/lemmatization, which can bias toward topical/style cues.

4. Baselines Missing: Stronger methods like RADAR [4], and commercial methods should be included to make the experiment rigorous.

5. Presentation Issues: The presentation could be better, e.g., there are repetitive texts all over the paper, missing any discussion or conclusion, etc.



References:

[1] Almost AI, Almost Human: The Challenge of Detecting AI-Polished Writing

[2] LLM-as-a-Coauthor: The Challenges of Detecting LLM-Human Mixcase

[3] IS CHATGPT INVOLVED IN TEXTS? MEASURE THE POLISH RATIO TO DETECT CHATGPT-GENERATED TEXT

[4] RADAR: Robust AI-Text Detection via Adversarial Learning

**Questions:**

Please see the weaknesses.

---

### Official Review · Reviewer_wVVJ · 2025-11-01

**Soundness:** 1
**Presentation:** 3
**Contribution:** 2
**Rating:** 2
**Confidence:** 5

**Summary:**

The paper proposes a framework for AI generated text detection, for a novel problem setting which is to distinguish human-written content and human-written, AI-edited content (hwAI-generated text).

**Strengths:**

The paper addresses a novel setting in the context of AI-generated text detection. Departing from existing frameworks that focus on distinguishing AI-generated text from human-written text, this new setup is refreshing and has potential for further exploration. The paper is overall well-written and easy to follow.

**Weaknesses:**

My first concern is about the justification of the new problem, namely distinguishing between AI-generated, human-rewritten, and AI-modified text. While I find the deviation from the traditional setup interesting, I am not sure what the potential use cases for this problem would be. Ideally, we would want to identify AI-generated text that has been modified by humans. I would appreciate it if the authors could provide concrete use cases to better justify this novel problem formulation.

Moreover, the proposed framework in the paper is based on prompt engineering, making it not robust and heavily dependent on prompt design. In addition, as part of the proposed method, the paper empirically curates a list of discriminative words; however, it remains unclear how accurate or comprehensive this list is without further justification or supporting evidence.

**Questions:**

Please refer to the weakness section.

---

### Note · Authors · 2025-11-29

**Comment:**

We thank the reviewers for the feedback and for pointing out the new references. However, we disagree with reviewers in several points:
- We disagree that our problem is not interesting. The second paragraph of our paper has provided examples of why this is a realistic and interesting problem. Detecting the human-AI collaboration is at least as interesting as (if not more than) detecting the purely AI-generated text.
- There seems a misunderstanding that our method only works for edits of word replacement. The long footnote on Page 2 has explained why our method works well for complex editing, where the key is that our vocabulary uses natural words without stemming and lemmatization.
- We don't think a method should be penalized due to it is "simple". Being simple does not mean being trivial. We view it as a blessing that a simple method can work so well.

We appreciate the reviewers' efforts, and we do find some of the comments useful. However, some of the required revisions are beyond the scope of this paper. Therefore, we decide to withdraw this submission.

**Withdrawal Confirmation:**

I have read and agree with the venue's withdrawal policy on behalf of myself and my co-authors.